# Is *Ficopomatus enigmaticus* (Annelida: Serpulidae) Becoming a Nuisance in Galveston Bay, Texas?

**Vanessa Fernández-Rodríguez** *[ID], **Laura Jurgens and Anja Schulze** [ID]

Department of Marine Biology, Texas A&M University at Galveston, P.O. Box 1675, Galveston, TX 77554, USA;
jurgensl@tamug.edu (L.J.); schulzea@tamug.edu (A.S.)
* Correspondence: vanessa.fernandezr@tamu.edu

**Abstract:** Herein, we report the recent proliferation of the non-native serpulid worm *Ficopomatus enigmaticus* in parts of Galveston Bay, Texas. Reef-like aggregations were first recovered from a settlement plate and a submerged brick at the Kemah Boardwalk Marina on 4 May 2023. By 20 May, similar growths were reported to us by the Kemah Boardwalk Marina on floating docks, boat hulls, propellers, and rudders. On June 8, nearby marinas remained unaffected. We confirmed the worms' species identities by noting their diagnostic characteristics.

**Keywords:** biofouling; invasive; estuary; suspension feeder; reef building; Gulf of Mexico; ecosystem engineer

Estuaries, especially those with major ports and heavy shipping traffic, are prone to invasion by non-native species [1–3]. Galveston Bay (GB), Texas, is a subtropical estuary located on the northern coast of the Gulf of Mexico. GB is home to three major ports: Houston, Galveston, and Texas City. The intense shipping traffic provides opportunities for the introduction of non-native species into the GB system through either ballast water or hull-fouling communities [4]. Another possible pathway for species introduction is aquaculture, which is of regional importance because commercial oyster farming only began in Texas in 2020 after a dramatic decline in natural oyster reefs that started in 2007 [5]. Non-native species are poorly documented in GB compared to other major shipping hubs in the US, which is partly due to a lack of baseline information about pre-industrial biological communities.

Herein, we report on the recent proliferation of the serpulid annelid *Ficopomatus enigmaticus* (Fauvel, 1923) in a part of GB. On 4 May 2023, we recovered a large aggregation of *F. enigmaticus* from a settlement panel at the Kemah Boardwalk Marina (KBM) and on a submerged brick, both of which were almost completely covered (Figure 1a). On 20 May 2023, we were contacted by the Marina General Manager at KBM, who reported large accumulations on floating docks, hulls, rudders, and the propellers of recreational boats (Figure 1b). On 8 June 2023, we inspected the Kemah Boardwalk Marina as well as the nearby Clear Lake Marine Center and the Seabrook Marina and Shipyard (Figure 2). Large aggregations of *F. enigmaticus* were only present at the KBM (Figure 1c,d). At the other locations, we only detected small numbers of individual tubes, although the environmental conditions were similar (Table 1). Although we previously recovered two additional *Ficopomatus* species (*F. uschakovi* and *F. miamiensis*) from settlement plates at the same location, we did not detect these species in the large aggregations. We have not yet identified any environmental conditions that might have triggered the sudden localized proliferation of *F. enigmaticus*.

*Ficopomatus enigmaticus* was first reported in Galveston Bay in 2003 [6]. In 2017, it was observed on settlement plates, together with the native *F. miamiensis* and the non-native *F. uschakovi* [7]. Between 2019 and 2022, it was regularly detected on settlement plates deployed seasonally in GB [8]. The worms were present at mesohaline stations (6–12 ppt in

the northern and central sites of GB and ~18 ppt in West Bay), but their numbers remained low [8]. The first reports of *F. enigmaticus* in the northern Gulf of Mexico emerged in the early 1950s in reference to Corpus Christi Bay [9] and Aransas Bay [10]. *Ficopomatus enigmaticus* has also been reported further east in Mississippi (1997) and Tampa Bay (2002) [6]. It does not seem to occur in significant numbers or form reef-like structures in any of these locations.

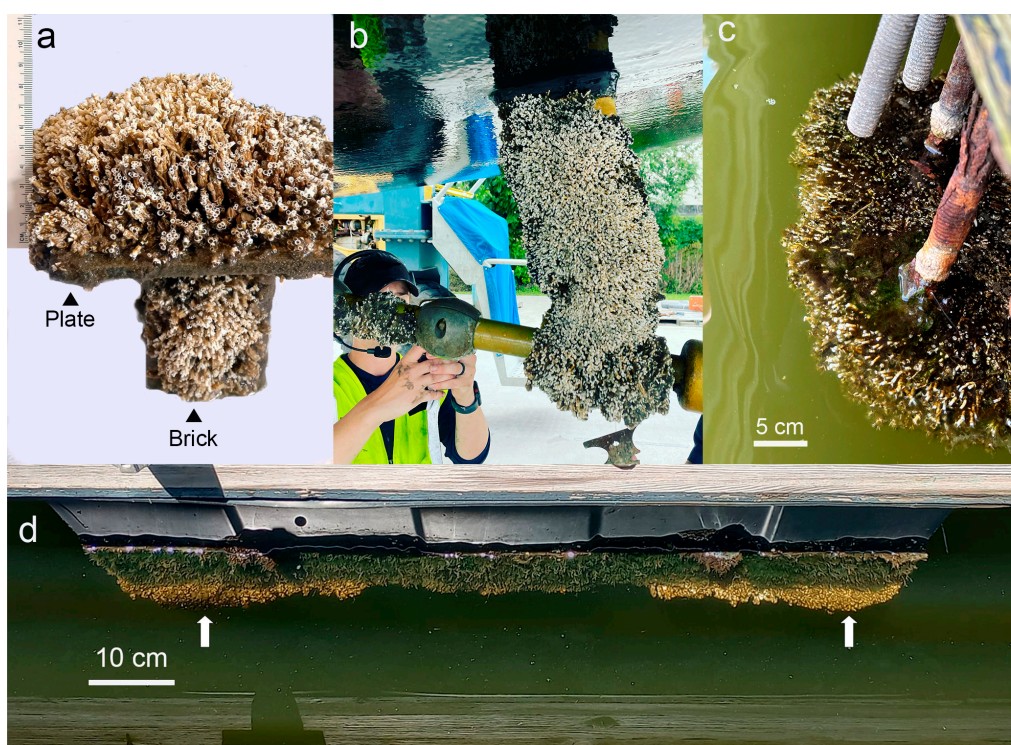

**Figure 1.** Massive aggregations of *Ficopomatus enigmaticus* in Kemah, Galveston Bay, Texas. (**a**) Colonization plate and brick recovered from Kemah Boardwalk Marina covered in tubes; (**b**) boat propeller; (**c**) patch of *F. enigmaticus* underneath a floating dock; (**d**) floating dock with *F. enigmaticus* (arrows point to the clusters). (**b**) was provided by the General Manager of Kemah Boardwalk Marina.

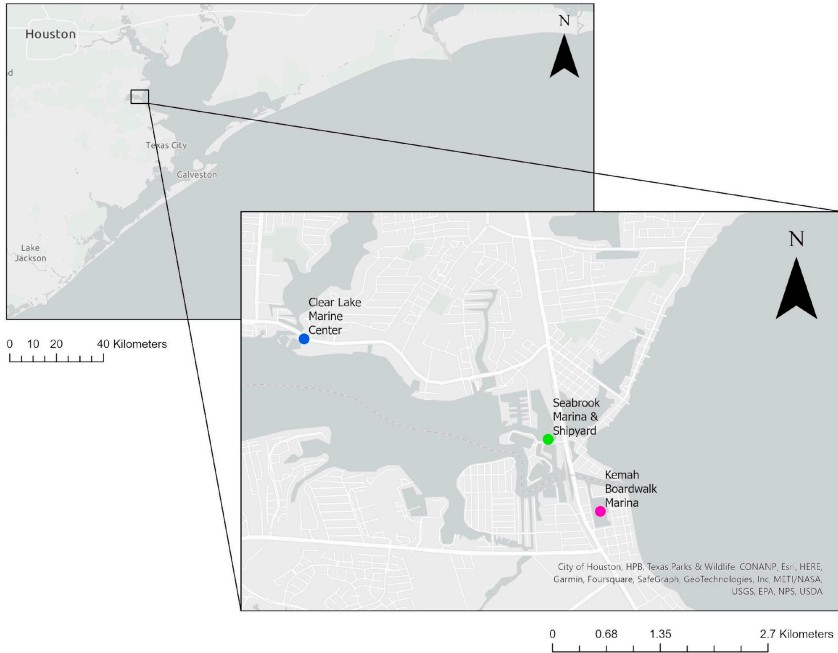

**Figure 2.** Map of the studied shipyard and marinas in Galveston Bay, Texas, USA.

**Table 1.** Water environmental conditions in the study locations. Measurements were taken at a depth of 1 m from the surface on 8 June 2023 using a YSI water quality instrument (Pro2030, Yellow Springs Instruments, Yellow Springs, OH, USA).

| Marina | Water Temperature (°C) | Salinity (ppt) | Dissolved Oxygen (mg/L) |
|---|---|---|---|
| Kemah Boardwalk Marina | 28.4 | 9.7 | 5.22 |
| Clear Lake Marine Center | 28.4 | 8.1 | 3.61 |
| Seabrook Marina and Shipyard | 28.6 | 10.2 | 5.29 |

*Ficopomatus enigmaticus* is known as as a worldwide invader in many temperate to subtropical coastal areas with brackish conditions [6]. Individual worms are generally 10–15 mm long, but worms as large as 39 mm have been reported [11]. Like other serpulid worms [7], *F. enigmaticus* secretes and inhabits calcareous tubes that exceed its body length by a factor of 2–3 and are 1–2 mm wide, possessing a flared anterior end [11]. Individual tubes can cluster together in reef-like structures that can be up to 7 m in diameter and grow to a height of 0.5 m [12]. Clusters can converge, with new recruitment over old aggregates, sometimes forming continuous reef-like structures up to 3 m thick [13]. We identified *F. enigmaticus* by referring to its diagnostic characteristics reported by Bastida-Zavala et al. [14]: a fig-shaped operculum (Figure 3a) with a horny plate covered with inwardly pointing dark brown to black spines (Figure 3b); well-developed thoracic membranes that are not fused dorsally (Figure 3a); and coarsely serrated chaetae in the collar (Figure 3c). Among the examined specimens, we found an individual with two opercula (Figure 3d), which were inserted on the right and left lobes, respectively. According to Bastida-Zavala et al. [14], only the genera *Apomatus* and *Filogranna*, and sometimes *Hydroides*, present duplicate opercula. We did not find previous reports of this phenomenon in previous studies; therefore, this is the first evidence of a specimen of *F. enigmaticus* with duplicate operculum.

The native range of *F. enigmaticus* is unclear. Although the species is often referred to as the "Australian tubeworm", Australia might not have been its geographic origin [15]. Massive invasions have occurred in the Mediterranean and Black Seas [11,16,17], the Baltic Sea [18], the North American west coast [19,20], the southwest Atlantic [21], New Zealand [22], and South Africa [23]. *Ficopomatus enigmaticus* has been reported in numerous other locations but does not always engage in reef building. Where it does form reefs, these reefs can have significant ecological and economic impacts. As an efficient suspension feeder, *F. enigmaticus* competes with other filter-feeding organisms over phytoplankton [24]. As an ecosystem engineer, it generally increases biodiversity by altering the community composition of the benthic fauna, including facilitating the establishment of other non-native species [25–28]. Another concern is the fouling of other organisms (e.g., oysters), which could alter their hydrodynamic properties and feeding efficiency. Tubeworm aggregates on artificial structures, such as boats, pilings, buoys, and bulkheads, can precipitate significant clean-up costs and pose a safety risk to swimmers. Tubeworm aggregations can also block pipes and sluice gates [29–31].

Whether the recent proliferation of *F. enigmaticus* in Kemah, Texas, will expand to other parts of GB or whether the worms will die back with seasonal weather changes remains to be seen. We are currently raising awareness among boaters and shipyard and marina operators to alert us of new occurrences. Genetic and ecological studies of the reef-building worms, and individual worms originating from different locations collected throughout all seasons in GB, are currently underway.

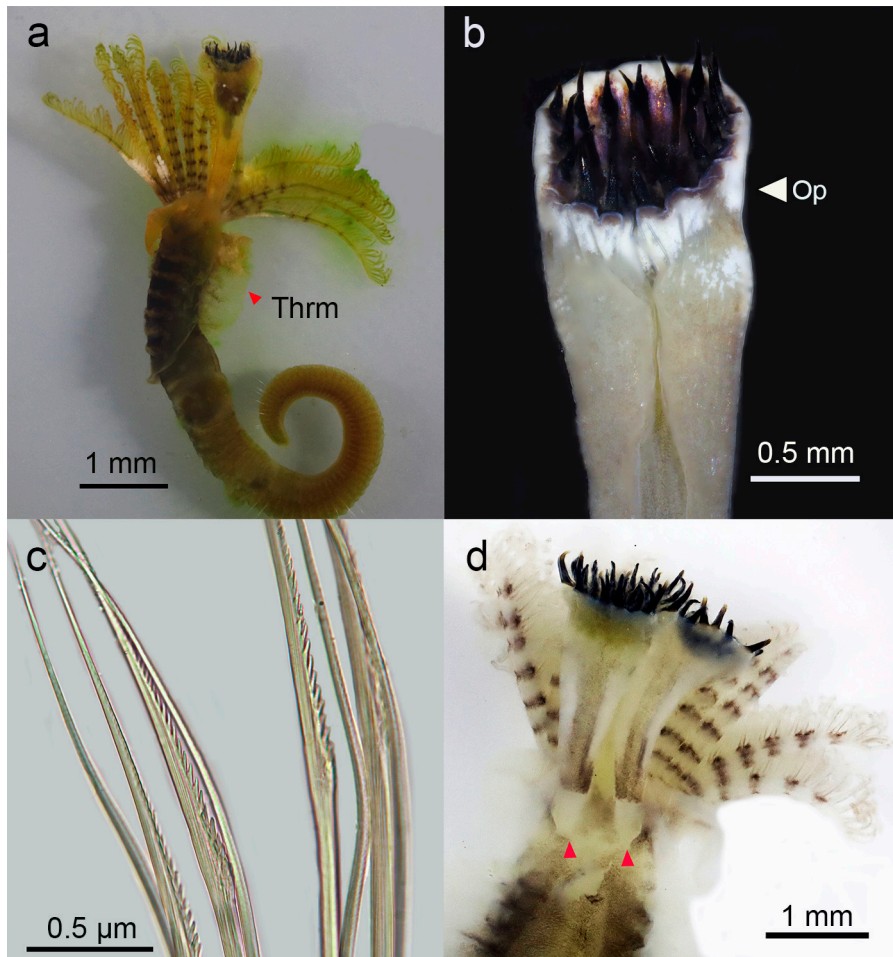

**Figure 3.** *Ficopomatus enigmaticus* (**a**) Complete body in lateral view (the red triangle indicates the thoracic membrane); (**b**) operculum in dorsal view; (**c**) coarsely serrated chaetae in the collar; (**d**) specimen of *F. enigmaticus* with duplicate operculum in dorsal view (the red triangles indicate the insertion points in the body). Abbreviations: operculum (Op); thoracic membrane (Thrm).

**Author Contributions:** Conceptualization, V.F.-R. and A.S.; methodology, V.F.-R. and A.S.; resources, L.J. and A.S.; data curation, V.F.-R.; writing—original draft preparation, A.S.; writing—review and editing, V.F.-R., L.J. and A.S.; visualization, V.F.-R.; supervision, A.S.; project administration, A.S. and L.J.; funding acquisition, A.S. and L.J. All authors have read and agreed to the published version of the manuscript.

**Funding:** This research was funded by a "Mentorship-through-Collaboration" grant (Marine Biology Department, Texas A&M University at Galveston) given to A.S. and L.J.

**Data Availability Statement:** All relevant data are included in the body of this manuscript.

**Acknowledgments:** We acknowledge the assistance of Brad Grace, General Manager at Kemah Boardwalk Marina, and the cooperation of the management of the Clear Lake Marine Center and the Seabrook Marina and Shipyard. We also thank Marissa J. Palmer for her assistance with the map.

**Conflicts of Interest:** The authors declare no conflict of interest.

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
