# Peer review of "Is Ficopomatus enigmaticus (Annelida: Serpulidae) Becoming a Nuisance in Galveston Bay, Texas?"

_diversity, doi:10.3390/d15070852_

Round 1

Reviewer 1 Report

I would suggest the authors finish their research (genetic studies and distribution) on the occurrence of F. enigmaticus in Kemah, TX, and publish a comprehensive study. However, even if the article would be published in this short form, could you please add some details on the factors that might have conducted to the proliferation of the worm after 2020? You have only mentioned aquaculture and shipping. Could also some peculiar changes in ecological conditions influence its spreading?

Author Response

Dear Reviewer #1

We would like to thank you for your feedback on this manuscript. Please see our answers or corrections below.

Authors’ answer: We are working on a comprehensive phylogeographic analysis to clarify spatial and temporal trends of Ficopomatus enigmaticus in Galveston Bay and identify potential sources of the worms, but data collection is still ongoing. We believe that reporting the recent proliferation in this short form is of interest because the population may change quickly before our comprehensive analysis is complete. We are also still collecting ecological data that will complement the discussion of F. enigmaticus spreading in Galveston Bay; and as we highlighted in the manuscript (lines 23 -25), non-native species are poorly documented in Galveston Bay. We inserted a sentence (lines 38-39) that we have not yet identified any environmental factors that may have contributed to the sudden localized proliferation. We have also re-written lines 89-91 to inform the readers that other aspects of this research are underway. Corrections in the revised manuscript are highlighted in blue.

Reviewer 2 Report

In a previous paper published in 2017, some colleagues find F. enigmaticus as well as F. uschakovi (both exotic/invader species) in fouling plates settled in Galveston Bay. Also, they find to F. miamiensis, native in Gulf of Mexico. You did found specimens of any of these species in your samples? Should be interesting known if other worms live in these F. enigmaticus aggregations o, by contrary, declare that are monospecific aggregations.

You quoted the 19 reference (page 3, line 68); however, the paper of Tovar-Hernández et al. (2022) is about two Ficopomatus species as invaders in Mexico, none is F. enigmaticus.

Maybe the journal allows it, but I don't like start a paragraph with a abbreviation. Is the case of F. enigmaticus (page 1, line 37).

I don't reviewed all references; however, I observed some mistakes, as "mpacts" in reference 10.

Author Response

Dear Reviewer #2

We would like to thank you for your feedback on this manuscript. Please see our answers or corrections below for each of your suggestions. Corrections in the revised manuscript are highlighted in blue.

In a previous paper published in 2017, some colleagues find F. enigmaticus as well as F. uschakovi (both exotic/invader species) in fouling plates settled in Galveston Bay. Also, they find to F. miamiensis, native in Gulf of Mexico. You did found specimens of any of these species in your samples? Should be interesting known if other worms live in these F. enigmaticus aggregations o, by contrary, declare that are monospecific aggregations.

Authors’ answer: In the paper of Bastida-Zavala et al., (2017), the authors deployed settlement panels and analyzed 12 worms of Ficopomatus enigmaticus from Galveston Bay. However, the authors did not specify the exact location within the bay. For the massive aggregations reported in this manuscript, we did not find specimens or signs (e.g., empty tubes) of F. miamiensis or F. uschakovi in the large aggregations. We inserted a sentence to clarify this in lines 36-39, and added the suggested reference in the introduction (lines 40-42), as follows:

  • Lines 36-39: “Although we previously recovered two additional Ficopomatus species ( uschakovi and F. miamiensis) from settlement plates at the same location, we have not detected these species in the large aggregations. We have not yet identified any environmental conditions that may have triggered the sudden localized proliferation of F. enigmaticus”.
  • Lines 40-42: “Later, [6] reported it on settlement plates in 2017, together with the native miamiensis and the non-native F. uschakovi”.

As we stated in the manuscript (lines 42-45), the constant presence of F. enigmaticus has been previously recorded within the bay in settlement panels; however, in none of those publications was a massive outbreak reported in any of the studied locations, nor was a massive outbreak of these worms previously found attached to floating docks, hulls, rudders, and propellers of recreational boats. Therefore, this is the first time in which a massive aggregation of F. enigmaticus is detected on those substrates, which open questions about its ecology, biology, and potential triggers that caused this phenomenon. The location at which this outbreak is reported (eastern Galveston Bay; see Fig 2 in the manuscript), is an area with a concentration of private shipyard marinas. The information presented in our manuscript will contribute to establishing a baseline in the monitoring of this non-native species in Galveston Bay. 

You quoted the 19 reference (page 3, line 68); however, the paper of Tovar-Hernández et al. (2022) is about two Ficopomatus species as invaders in Mexico, none is F. enigmaticus.

Authors’ answer: The mentioned reference was deleted since the publication of Tovar-Hernández et al. (2022) is not about F. enigmaticus.

Maybe the journal allows it, but I don't like start a paragraph with a abbreviation. Is the case of F. enigmaticus (page 1, line 37).

Authors’ answer: Corrected, with our thanks. See lines 40 (line 37 in the previous version) and 53.

I don't reviewed all references; however, I observed some mistakes, as "mpacts" in reference 10.

Authors’ answer: The complete list of references was revised. Please note that after deleting Tovar-Hernández et al. (2022) and adding Bastida-Zavala et al., (2017), the reference numbers in the revised version changed.

Typing mistake in reference 11 (#10 in the previous manuscript version) was corrected. We did not find any other mistakes on the list. Please note that title in reference 16 (#15 in the previous manuscript version) is written in the same way it appeared in the publication (https://doi.org/10.1111/j.1439-0485.1996.tb00489.x), and this does not correspond to a mistake in our reference list.      

Reviewer 3 Report

General Comments:

The paper reports on a recent proliferation of the non-native serpulid worm Ficopomatus enigmaticus in parts of Galveston Bay, TX. The information provided could be useful in establishing a baseline information about non-native species in Galveston Bay and in the implementation of mitigation and control actions that must follow the assessment of the current environmental status.

The title is informative and a good reflection of the content. The Abstract cover all salient points of the study. The introduction part clearly states the problem. Also, the study is placed in an appropriate context by referring to relevant previous works. Results are presented clearly. The figures are clear, informative and self-explanatory. Overall, the writing is very good.

Specific Comments:

I have no comments or suggestions to be made for the ms. In conclusion, a sound and well executed paper! Therefore, my recommendation is to accept it in the present form.

Author Response

Dear Reviewer # 3. We appreciate your positive comments about our manuscript.